# Lysophosphatidic Acid Receptor 5 (LPA_5_) Knockout Ameliorates the Neuroinflammatory Response In Vivo and Modifies the Inflammatory and Metabolic Landscape of Primary Microglia In Vitro

**DOI:** 10.3390/cells11071071

**Published:** 2022-03-22

**Authors:** Lisha Joshi, Ioanna Plastira, Eva Bernhart, Helga Reicher, Zhanat Koshenov, Wolfgang F. Graier, Nemanja Vujic, Dagmar Kratky, Richard Rivera, Jerold Chun, Wolfgang Sattler

**Affiliations:** 1Gottfried Schatz Research Center, Division of Molecular Biology and Biochemistry, Medical University of Graz, 8010 Graz, Austria; lisha.joshi@medunigraz.at (L.J.); ioanna.plastira@medunigraz.at (I.P.); eva.bernhart@medunigraz.at (E.B.); helga.reicher@medunigraz.at (H.R.); zhanat.kostenov@medunigraz.at (Z.K.); wolfgang.graier@medunigraz.at (W.F.G.); nemanja.vujic@medunigraz.at (N.V.); dagmar.kratky@medunigraz.at (D.K.); 2BioTechMed-Graz, 8010 Graz, Austria; 3Translational Neuroscience Initiative, Sanford Burnham Prebys Medical Discovery Institute, La Jolla, CA 92037, USA; rrivera@sbpdiscovery.org (R.R.); jchun@sbpdiscovery.org (J.C.)

**Keywords:** chemokines, cytokines, nitric oxide, endotoxin, immunometabolism, lysophospholipids

## Abstract

Systemic inflammation induces alterations in the finely tuned micromilieu of the brain that is continuously monitored by microglia. In the CNS, these changes include increased synthesis of the bioactive lipid lysophosphatidic acid (LPA), a ligand for the six members of the LPA receptor family (LPA_1-6_). In mouse and human microglia, LPA_5_ belongs to a set of receptors that cooperatively detect danger signals in the brain. Engagement of LPA_5_ by LPA polarizes microglia toward a pro-inflammatory phenotype. Therefore, we studied the consequences of global LPA_5_ knockout (^-/-^) on neuroinflammatory parameters in a mouse endotoxemia model and in primary microglia exposed to LPA in vitro. A single endotoxin injection (5 mg/kg body weight) resulted in lower circulating concentrations of TNFα and IL-1β and significantly reduced gene expression of IL-6 and CXCL2 in the brain of LPS-injected LPA_5_^-/-^ mice. LPA_5_ deficiency improved sickness behavior and energy deficits produced by low-dose (1.4 mg LPS/kg body weight) chronic LPS treatment. LPA_5_^-/-^ microglia secreted lower concentrations of pro-inflammatory cyto-/chemokines in response to LPA and showed higher maximal mitochondrial respiration under basal and LPA-activated conditions, further accompanied by lower lactate release, decreased NADPH and GSH synthesis, and inhibited NO production. Collectively, our data suggest that LPA_5_ promotes neuroinflammation by transmiting pro-inflammatory signals during endotoxemia through microglial activation induced by LPA.

## 1. Introduction

Normal brain function depends on a balanced set of diverse lipid species, which have structural as well as signaling roles. Within the body, the brain has the second highest lipid content and diversity after adipose tissue [1]. High-resolution mass spectrometry-based lipidomic approaches revealed a distinct lipid signature of the brain that is substantially different from non-neural tissues, such as muscle and kidney [2]. Bioactive signaling lipids transmit their function by binding to specific receptors [3,4,5] and activating the downstream pathways [5,6]. In the periphery and the central nervous system (CNS), lysophosphatidic acid (LPA) species represent an important subclass of bioactive signaling lipids [6]. Signaling is determined by LPA receptor-triggered pathways in which six LPA receptors (LPA_1-6_) couple to distinct classes of heterotrimeric G proteins, leading to a variety of cellular activities [7,8]. In the CNS, the LPA/LPAR axis plays a pivotal role during neurodevelopment and homeostasis by modulating neurotransmission, synaptic plasticity, enzyme function, gene expression, and (neuro)inflammatory reactions [6].

Immune activation in the periphery affects the function of the CNS and is associated with an increased risk of developing neuropsychiatric and neurodegenerative diseases [9]. Microglia play an important role in this scenario since there is growing evidence that microglia activation and neuroinflammation can be induced by systemic inflammatory events [10,11]. This is ascribed to their ability to continuously monitor changes in the brain microenvironment through a specifically enriched set of surface receptors termed the “microglia sensome” [12]. Early studies had identified LPA receptor gene expression on rodent microglia [13], while recently validated mouse and human microglia RNAseq data sets define the microglia core sensome as a key set of 57 genes conserved between the mouse and human sensome [14] including LPA_5_ in mouse and human microglia (along with additional expression of LPA_6_ in humans) associated with the core sensome [14].

LPA_5_ [15] and LPA_6_ [16,17] are highly expressed in BV-2 cells and primary murine microglia [18,19], although LPA receptor expression in the brain is subject to developmental regulation [6,20] and depends on the genetic background of the animal model used [21]. LPA_5_ is considered to be a driving factor for acute and chronic ischemic injuries in the mouse transient middle cerebral artery occlusion (tMCAO) stroke model, and the LPA_5_ antagonist TCLPA5 has been shown to confer acute and long-term protection in this injury model [22,23]. These findings were ascribed to a pathogenic role of LPA_5_ closely associated with microglia activation in injured brains [22], likely via RAGE-dependent pathways [24]. Targeted deletion of LPA_5_ identified a novel role for this receptor in mechanically or chemically induced murine neuropathic pain models [25,26]. Consistently, the LPA_5_ inhibitor AS2717638 ameliorated mechanical allodynia and thermal hyperalgesia in rodent models of neuropathic pain [27]. In microglia, LPA_5_ induces a polarization program toward a neurotoxic phenotype, since genetic (siRNA) [22] and pharmacological (TCLPA5, Cpd3, AS2717638) approaches revealed that inhibition of this receptor attenuates the neuroinflammatory output of the cells [28,29].

LPA signaling plays an important role in microglia polarization [18,30], the neuroinflammatory response [31], and a shift in cellular metabolic pathways [32,33]. To monitor the role of LPA_5_ in endotoxemia in the absence of pharmacological antagonists (and potentially associated off-target effects), we used a global LPA_5_ knockout mouse model. Using this mouse model, we studied the peripheral and central inflammatory response in an endotoxemia model utilizing wildtype (wt) and LPA_5_-deficient (^-/-^) mice. In vitro, we investigated the impact of LPA on cyto-/chemokine secretion and basic immunometabolic parameters (mitochondrial function, lactate, NADPH, GSH, and nitric oxide synthesis) in microglia isolated from newborn wt and LPA_5_^-/-^ mice.

## 2. Materials and Methods

### 2.1. Animals

Mice were housed and bred in a clean environment and a 12 h/12 h light–dark cycle with chow diet and water access ad libitum. All animal experiments were approved by the Austrian Federal Ministry of Education, Science and Research (BMWF-66.010/0067-V/3b/2018 and 2020_0.547.884). All measures were taken to minimize animal suffering and distress.

The genotyping protocols for LPA_5_^-/-^ genetically modified mice have been described previously [25]. Briefly, homozygous LPA_5_-null mutant mice were generated by targeted deletion of the LPA_5_ gene to eliminate and replace most of the LPA_5_ coding region in C57BL/6J mice. Heterozygous mice were mated to each other to obtain wildtype (wt; +/+) and null mutant offspring (^-/-^). Genotypes were confirmed by PCR genotyping using the following primers:GFP Int Rev, 5-GTGGTGCAGATGAACTTCAGG- 3;92GTFor, 5-CAGAGTCTGTATTGCCACCAG- 3; and92GT Rev, 5-GTCCACGTTGATGAGCATCAG- 3.

Male wt and LPA_5_^-/-^ mice aged 12–16 weeks weighing 20–30 g were injected i.p. with PBS or LPS (5 mg/kg body weight in PBS). Twenty-four hours post injection, the animals were euthanized, perfused, and brains were collected in QIAzol Lysis Reagent (Qiagen, Hilden, Germany) for RNA isolation.

Blood (200–300 µL) was isolated by cardiac puncture. The tubes containing the blood samples were kept at room temperature for 1 h and then centrifuged at 5000× *g* for 10 min. The clear supernatant (serum) was collected, diluted 1:10 and used for ELISA analyses.

### 2.2. Indirect Calorimetry (Metabolic Cage Monitoring)

LPA_5_^-/-^ and wt mice were individually housed in PhenoMaster cages (TSE Systems, Bad Homburg, Germany). We determined energy intake and expenditure as well as ambulatory movements in the mice over 5 days before and for 4 days during chronic LPS i.p. application (1.4 mg/kg body weight in PBS every 24 h). We chose this chronic low-dose regimen to observe animal behavior over an extended period of time, which is not possible in the acute high-dose LPS model (5 mg/kg body weight) since animal suffering increases at time points > 1 d post LPS application [34]. A comparable chronic treatment regimen (four daily LPS injections of 1 mg/kg) was shown to induce global microglia activation in C57BL/6 mice [35]. Oxygen consumption (VO_2_) and carbon dioxide production (VCO_2_) were simultaneously measured every 15 min by indirect gas calorimetry.

Since food consumption from the built-in food containers (which are localized above the light beams that monitor locomotion) may be too challenging for some animals after the LPS injections, we decided to place food pellets directly in the cage and measure food consumption manually. During the adaption phase, cumulative food consumption was divided by the days of the adaption phase to measure average food consumption. During the LPS treatment, food consumption was measured manually every day just before the LPS injection.

### 2.3. RT-qPCR Analysis

We isolated total RNA from the brain with the RNeasy Lipid Tissue Mini Kit (QIAGEN, Hilden, Germany) according to the manufacturer’s protocol. Total RNA was quantitated using NanoDrop (Thermo Fisher Scientific, Waltham, MA, USA) and reverse-transcribed using the SuperScript^®^ III reverse transcription kit (Invitrogen, Waltham, MA, USA). Quantitative real-time PCR (qPCR) using the QuantifastTM SYBR^®^ Green PCR kit (QIAGEN, Hilden, Germany) was performed on the Applied Biosystems 7900HT fast real-time PCR system. Gene expression was normalized to the expression of hypoxanthineguanine phosphoribosyltransferase (HPRT) as housekeeping gene. Expression profiles and associated statistical parameters were calculated using the 2^−ΔΔCT^ method. Primer sequences are listed in Table 1.

### 2.4. Primary Microglia Cultures

To isolate primary murine microglia from the cortices of newborn (P0–P4) wt and LPA_5_^-/-^ mice, we dissected brain cortices from the entire brain, and the meninges were removed and cut into small pieces with scissors. Tissues were trypsinized (0.1% trypsin, 25 min, 37 °C, 5% CO_2_), and centrifuged at 1700 rpm for 7 min. The supernatant was aspirated and the pellet was suspended in DMEM. This cell suspension was cultured in poly-D-lysine (PDL; 5 μg/mL)-coated 75 cm^2^ tissue culture flasks (4 brains per flask) in DMEM supplemented with 15% FCS, 1% penicillin, 1% streptomycin, and 1% L-glutamine. After cultivation of the cells for another 10 to 14 days, we removed microglia from the mixed glia cell cultures by vigorously tapping the culture flasks on the bench top. Microglia were then seeded onto PDL-coated cell culture plates for further use.

### 2.5. LPA Treatment

The aqueous LPA (1-Oleyl-2-hydroxy-sn-glycero-3-phosphate; Sigma-Aldrich, St. Louis, MO, USA; Cat. L7260) stock solution (5 mM) was aliquoted and stored at −70 °C. Fresh aliquots were used for the experiments. Primary microglia were plated out in 12- or 24-well plates and allowed to adhere for 2 days. Before treatments, cells were incubated in serum-free DMEM overnight. The following day, fresh, serum-free medium was added, followed by the addition of LPA.

### 2.6. Seahorse XF Analyzer Respiratory Assay

Cells (6 × 10^4^ cells per well) were seeded in Seahorse XFe96 FluxPaks for metabolic analysis with an extracellular flux analyzer XF96 (Seahorse, Agilent, Santa Clara, CA, USA). The sensor cartridge was hydrated in a 37 °C non-CO_2_ incubator one day before the experiment. Cells were serum-starved overnight, treated with LPA at the indicated concentrations for the given time periods, washed, and incubated with the appropriate assay medium for 1 h in a 37 °C non-CO_2_ incubator according to the manufacturer’s instructions. Cellular oxygen consumption rate (OCR) was determined using the XF Cell Mito Stress Test (Agilent). Optimized stressor concentrations were added as follows: 2 μM oligomycin (complex V inhibitor), 1.75 μM cyanide p-trifluoromethoxy-phenylhydrazone (FCCP; proton gradient disruption), and 2.5 μM antimycin A (inhibitor of complex I and III). OCR was normalized to protein concentrations, and data from 3 independent experiments are shown.

### 2.7. ELISA

Primary cells (5 × 10^5^ per well) were seeded onto 6-well plates, serum-starved overnight, and then treated with LPA for the indicated time periods. Thereafter, the supernatant was collected and stored at −80 °C until further use. Murine ELISA development kits (Peprotech, Cranbury, NJ, USA) were used to determine concentrations of cytokines (IL-1β, TNFα, IL-6) and chemokines (CCL5 (RANTES), CXCL2 (MIP-2), and CXCL10 (IP-10)) using external standard curves.

### 2.8. Lactate Measurement

Lactate content in the supernatant was measured using the EnzyChrom™ Glycolysis Assay Kit (ENZO Life Sciences, Lausen, Switzerland) according to the manufacturer’s protocol. Briefly, primary microglia (96-well plate, 6 × 10^4^ cells per well) were allowed to adhere, then incubated in serum-free medium and treated with LPA for indicated time periods. At the end of the treatment, the supernatant was collected and treated with the enzyme mix under brief shaking. Optical density was measured at 565 nm to quantify the lactate content.

### 2.9. NADPH/NADP Assay

Nicotinamide nucleotides were assayed using the NADP/NADPH assay kit (Abcam, Cambridge, UK) according to the manufacturer’s instructions. Primary microglia were seeded onto PDL-coated 12-well plates at a density of 5 × 10^5^ per well and serum-starved overnight prior to the experiments. Cells were treated with the given concentrations of LPA for the indicated time periods, after which the medium was removed, cells were washed twice with ice-cold PBS, and NADP/NADPH were extracted with the extraction buffer. The samples were deproteinized using 10 kDa Spin Columns (Abcam, Cambridge, UK) before performing the assay. An aliquot of the sample was used to measure total NADPt (NADP and NADPH). Another part of the sample was heated at 60 °C for 30 min to decompose NADP for NADPH measurement. Ten µL of the sample were mixed with NADP cycling mix to convert NADP to NADPH. Thereafter, 10 µL of NADPH developer were added into each well, mixed, and incubated at room temperature for 1–4 h. Multiple readings were taken at OD 450 nm. The ratio of NADPH/NADP was calculated as follows: NADPH/NADP ratio = NADPH/(NADPt − NADPH).

### 2.10. Glutathione Assay

Primary microglia (5 × 10^4^ per well) were seeded overnight in clear-bottom black 96-well plates to allow cells to adhere, incubated in serum-free medium, and treated with LPA (1 or 5 µM) for the indicated time periods. To measure intracellular glutathione content, cells were incubated with the GSH-Glo™ reagent (GSH-Glo™ Glutathione Assay Kit; Promega Corporation, Madison, WI, USA) for 30 min. After addition of the luciferin detection reagent, luminescence was measured to quantify the glutathione content according to the manufacturer’s protocol.

### 2.11. Determination of Nitric Oxide (NO)

Total nitrate content was measured in the supernatant of cells incubated with the indicated compounds in serum-free medium using the total nitric oxide assay kit (ENZO Life Sciences, Lausen, Switzerland) according to the manufacturer’s protocol. This assay detects a colored azo-dye product after the enzymatic conversion of nitrate to nitrite by nitrate reductase, followed by the Griess reaction. Nitrite concentrations in the samples were calculated by a standard curve in the range of 0–100 μM using nitrate as standard.

### 2.12. Statistical Analysis

Data are presented as mean ± SEM of at least 3 independent experiments (performed in triplicate), unless otherwise stated. Statistical analyses were performed using GraphPad Prism6 software. Significance was determined by unpaired Student’s *t*-test (two groups) or two-way ANOVA followed by Bonferroni correction (>two groups). Values of *p* < 0.05 were considered significant.

## 3. Results

### 3.1. Endotoxemia Is Reduced in LPA_5_^-/-^ Mice

In vivo, we investigated a potentially protective role of global LPA_5_ deficiency in an LPS-induced endotoxemia mouse model. Adopting a previously published protocol [36], wt and LPA_5_^-/-^ mice were injected i.p. with LPS (5 mg/kg body weight) and sacrificed 24 h later. Cyto-/chemokine concentrations and gene expression were determined in serum and in brain homogenates, respectively. Serum ELISA measurements revealed that LPS administration significantly increased TNFα, IL6, and IL-1β concentrations (Figure 1A–C). In LPA_5_^-/-^ mice, this increase was significantly lower for TNFα and IL-1β, while IL-6 concentrations remained unaffected (Figure 1A–C).

Following brain RNA isolation, gene expression of TNFα, IL-6, IL-1β, CXCL10, CXCL2, CCL5, iNOS, and Arg1 was analyzed by qPCR. In both genotypes, cyto- and chemokine expression levels were significantly enhanced in response to LPS when compared to vehicle (PBS)-injected animals (Figure 2A–F). iNOS (M1 marker) and Arg-1 (M2 marker) were upregulated in response to LPS; however, in wt animals, this effect was statistically not significant (Figure 2G,H). In LPA_5_^-/-^ mice, the increase of cyto- and chemokine mRNA expression was consistently lower as compared to wt (Figure 2A–F). These observations reached statistical significance for IL-6 and CXCL2.

### 3.2. Improved Metabolic Performance of LPA_5_^-/-^ Mice after Short-Term, Low-Dose LPS Treatment

To monitor potential differences in animal behavior and energy expenditure, wt and LPA_5_^-/-^ mice were housed in metabolic cages and subjected to a low-dose, chronic LPS regimen (1.4 mg LPS/kg body weight every 24 h for 4 d). During the preceding 5 d adaptation phase, wt or LPA_5_^-/-^ mice with ad libitum access to food and water were individually kept in metabolic cages. Under these basal conditions, feeding behavior, locomotion during the night cycle, respiratory exchange ratio (RER) and energy expenditure (EE) during the night cycle of wt mice were not significantly different from LPA_5_^-/-^ animals (Appendix A). Locomotion and energy expenditure during the day cycle was slightly lower (*p* < 0.05) for LPA_5_^-/-^ mice (Appendix A).

In comparison to basal conditions, animals of both genotypes exhibited classical signs of sickness behavior 24 h after the first LPS application (Figure 3). This is reflected by decreased water and food intake (Figure 3A,B), locomotion (Figure 3C,D), RER (Figure 3E,F) and EE (Figure 3G,H). However, in contrast to wt mice, several parameters of sickness behavior were significantly less pronounced in LPA_5_^-/-^ animals during the early acute inflammatory phase 24 h after the first LPS dose. This was reflected by higher water (night cycle) and food consumption, locomotor activity, as well as RER (night cycle) in comparison to LPS-injected wt animals (Figure 3A,B,D,F). The corresponding data for the entire 96 h monitoring period are shown in Appendix A and indicate that the animals partially recovered from LPS-induced sickness behavior (despite the consecutive injection). Most of the metabolic parameters (except total food intake and RER during the night cycle; Appendix A) were comparable between wt and LPA_5_^-/-^ mice.

### 3.3. LPA_5_ Regulates LPA-Induced Secretion of Cyto-/Chemokines in Primary Microglia

In response to acute or chronic endotoxemia, LPA levels and gene expression of autotaxin (ATX) and several LPA receptors (including LPA_5_) are upregulated in mouse brain homogenates [31]. Since LPA provides an induction signal for the transition of microglia toward a pro-inflammatory phenotype, we tested the hypothesis that LPA_5_ deficiency might affect this phenotypic switch. Indeed, analysis of cyto-/chemokine secretion in wt and LPA_5_^-/-^ microglia revealed remarkable differences between the two genotypes. LPA increased the concentrations of TNFα, IL-6, and IL-1β (Figure 4A–C) and the chemokines CXCL10, CXCL2, and CCL5 (Figure 4D–F) at one or more time points in wt microglia. In LPA_5_^-/-^ cells, this pro-inflammatory response was attenuated, with significantly reduced secretion of TNFα and IL-6 (Figure 4A,B). LPA-induced effects on IL-1β (Figure 4C) and chemokine secretion (Figure 4D–F) were virtually absent in LPA_5_^-/-^ cells.

To account for potential effects mediated by other LPA receptors (in particular LPA_6_ that is highly expressed by primary microglia [18]) or potential LPA loss during sample preparation [37], all in vitro experiments were performed also with 5 µM LPA. The higher LPA concentration was chosen to account for the lower LPA affinity of LPA_6_ [17]. Similar results for cyto-/chemokine secretion were obtained in response to 5 µM (Appendix A).

### 3.4. LPA_5_^-/-^ Microglia Have Higher Mitochondrial Capacity as Compared to wt Cells

Microglia, like peripheral immune cells, are able to utilize different energy metabolites to immediately adapt to chemical alterations in the local microenvironment [38]. Earlier studies from our group have indicated that LPA alters the metabolic profile of the mouse BV-2 microglia cell line to a glycolytic phenotype via an AKT/mTOR/HIF1α-dependent pathway [32] and drives them toward a pro-inflammatory phenotype via LPA_5_ [29,36]. To gain insight into whether the core sensome member LPA_5_ contributes to metabolic plasticity in microglia, we monitored basic metabolic parameters and inflammatory output in response to LPA in wt and LPA_5_^-/-^ cells.

In the first set of experiments, we examined mitochondrial function in real time using the Seahorse XF Cell Mito Stress Test. During Seahorse flux analysis, we compared mitochondrial function between wt and LPA_5_^-/-^ cells under basal (PBS) and LPA-activated conditions and assessed changes in the oxygen consumption rate (OCR) after treatment with an ATP synthase inhibitor (oligomycin), H^+^ ionophore (FCCP), and under electron-transport chain inhibition (rotenone and antimycin A). These experiments revealed that, under basal conditions, LPA_5_^-/-^ microglia showed higher OCR after 2 and 24 h in comparison to wt cells (Figure 5A,B). In response to LPA (1 µM, 2 h), maximal respiration (Figure 5E) was significantly higher in LPA_5_^-/-^ microglia as compared to wt cells, whereas basal respiration (Figure 5C), ATP production (Figure 5D), and spare respiratory capacity (Figure 5F) showed an upward trend. The increase in the latter two parameters was sustained up to 24 h in LPA_5_^-/-^ cells (Figure 5E,F). Qualitatively and quantitatively comparable observations were made during Seahorse analysis of primary cells cultured in the presence of 5 µM LPA (Appendix A).

### 3.5. LPA_5_ Deletion Attenuates LPA-Induced Lactate, NADPH, GSH, and NO Synthesis

In response to pro-inflammatory stimuli, metabolism of microglia shifts from OXPHOS to aerobic glycolysis [39]. To investigate potential differences in metabolic rewiring in LPA-polarized wt and LPA_5_^-/-^ cells, we quantified the levels of extracellular lactate, the major end product of aerobic glycolysis. These analyses revealed that lactate secretion by wt microglia was significantly increased by LPA (0.07 vs. 0.23 and 0.38 vs. 0.52 mM; control vs. LPA at 2 h and 24 h, respectively; Figure 6A). In LPA_5_^-/-^ cells, this rise in LPA-induced lactate secretion was virtually absent, indicating an important role for LPA_5_ during maintenance of metabolic plasticity.

To get an indication about the cellular redox status of LPA-treated wt and LPA_5_^-/-^ microglia, we analyzed the intracellular NADPH/NADP ratio, reduced glutathione (GSH) content, and nitric oxide (NO) release. The rationale for these analyses is based on earlier observations from our group [32], in which we showed that LPA treatment of BV-2 cells induces phosphorylation of nuclear factor erythroid 2-related factor 2 (Nrf2). Activation of the Nrf2 pathway transcriptionally regulates several key enzymes involved in the antioxidant response, including glucose-6-phopshate dehydrogenase (G6PD) and glutamate cysteine ligase subunits that catalyze the first step of GSH synthesis. Activation of G6PD activates the pentose phosphate cycle, which (in the oxidative branch) generates NADPH as an indispensable co-factor for NO synthesis via inducible NO synthase (iNOS). The NADPH/NADP ratio in wt cells was increased by 5-fold in response to a 24 h LPA exposure, whereas this response was significantly attenuated in LPA_5_^-/-^ cells (Figure 6B). In response to LPA, wt microglia time-dependently increased their intracellular GSH content. In contrast, under basal conditions, the GSH content was slightly lower (statistically not significant) in LPA5^-/-^ cells, whereas the LPA response of intracellular GSH levels was almost absent at all time points investigated (Figure 6C). As previously reported [40], LPA treatment increased NO concentrations in the cellular supernatant leading to a 2.8-fold increase after 24 h. This response was not observed in LPA_5_^-/-^ cells (Figure 6D). Comparable results were obtained with cells exposed to 5 µM LPA (Appendix A).

## 4. Discussion

Although it is not entirely clear under which conditions peripheral LPS crosses the blood–brain barrier (BBB) to trigger neuronal damage [41,42], there is a consensus that peripherally induced endotoxemia has the potential to cause neuroinflammation in experimental animal models and the human organism [43]. This results in microglia and astrocyte activation, memory loss, as well as destruction of synapses and apoptosis of neurons [44]. We have previously demonstrated that pharmacological interference with the ATX/LPA/LPA_5_ axis attenuates LPS-induced neuroinflammation in vivo and in a microglia cell line in vitro [36]. Here, we confirm and extend these findings to LPA_5_^-/-^ mice and primary LPA_5_^-/-^ microglia, thereby eliminating concerns regarding potential off-target effects of synthetic LPA_5_ antagonists. In particular, this global knockout model enabled us to directly demonstrate the involvement of LPA_5_ during the LPS-mediated peripheral and central inflammatory response, and a certain (short-term) contribution to energy expenditure and sickness behavior. In addition, the pro-inflammatory response of microglia towards LPA was less pronounced in LPA_5_^-/-^ microglia, a fact that might be partially attributed to the different immunometabolic phenotype induced in the knockout cells. A question arising from the present study is clinical translatability. As a lipid-activated receptor belonging to the GPCR family, LPA_5_ (among the other members of the receptor family) clearly qualifies as a druggable target in the CNS [45]. GPCRs are of high interest as pharmacological targets, since they are involved in pathophysiology, and druggable sites within this receptor class are accessible at the extracellular leaflet. Consequently, approximately 35% of FDA-approved drugs act on GPCRs [46]. Although drug delivery to the brain is restricted by the BBB [47], the situation for the LPA5 antagonist AS2717638 is encouraging since it accumulates in rat brain and displays neuroprotective action [27]. However, considering known differences between mouse and human immunology, findings in murine models must not necessarily reproduce in the human system [48].

Although Banks and Robinson have reported minimal BBB penetration of i.v.-injected ^125^I-LPS [41], Vargas-Caraveo suggested that LPS may be transported across the BBB via a lipoprotein-mediated pathway [42]. There is also evidence that LPS associates with Aβ_1-40/42_ within amyloid plaques and around vessels in brains of patients suffering from Alzheimer’s disease [49]. In addition, i.p. application of LPS provokes several features of sepsis and potentiates peripheral synthesis of cytokines and chemokines. Among these, TNFα has been shown to induce neuroinflammation via the TNF receptor 1 signaling pathway [50]. Additionally, this LPS transcytosis-independent pathway causes elevated gene expression of iNOS and TNFα in the brain [50], in accordance with data obtained in the present study (Figure 1 and Figure 2). Thus, reduced peripheral TNFα synthesis (Figure 1) would be expected to attenuate the neuroinflammatory response. Rat and human brain microvascular endothelial cells express TLR2, TLR3, TLR4, and TLR6 [51]. LPS interaction with TLR4 on these essentially non-immune cells initiates an inflammatory response by inducing IL-1β, IL-18, IL-6, and TNFα production [52]. In response to peripheral LPS, LPA concentrations in mouse brain and serum increase significantly and could amplify the inflammatory response [31]. Of note, LPA stimulates CD14 transcription and translation [53] and could, via activation of this TLR-4-associated coreceptor, enhance the LPS-mediated inflammatory response. Thus, multiple pathways initiated at the periphery may converge at the BBB, leading to an inflammatory microglia phenotype in the CNS in response to peripheral LPS.

Microglia play a central role in the initiation of neuroinflammation by surveying the chemical composition of the environment. Of relevance for the present study, LPA_5_ was identified as a core sensome member in mouse and human microglia [14]. As an extracellular, ligand-activated receptor almost exclusively expressed by microglia/macrophages (https://www.brainrnaseq.org/ (accessed on 7 February 2022); [54]), it is located in a strategic position to detect pathological alterations in the extracellular milieu and convey information to induce a phenotypic transition of microglia. Consequently, LPA_5_ was shown to play a disease-amplifying role in stroke [22,23,55], neuropathic pain [25,27,56], itching sensation [57], neuroinflammation [36], and during microglia polarization toward a neurotoxic phenotype [29,36]. Our group has previously shown that pharmacological antagonism of ATX (PF8380) and LPA_5_ (AS2717638) downregulated gene and protein expression of several pro-inflammatory markers in brains of LPS-injected C57BL/6 mice [36]. In line with this, results of the present study demonstrated reduced cyto-/chemokine gene expression in brains of LPS-injected LPA_5_^-/-^ animals (Figure 2), thereby replicating findings obtained with the LPA_5_ antagonist AS2717638. These data validate and re-confirm an important role of this lipid-activated receptor during induction of neuroinflammatory symptoms. Whether these observations are due to downregulation of TLR4 expression and subsequent attenuation of the inflammatory response (as observed with AS2717638; [36]) was not experimentally addressed in the present study.

In vivo energy metabolism was comparable between wt and LPA_5_^-/-^ mice under basal conditions (Appendix A). In the endotoxemia model, global LPA_5_ deficiency provided protection against LPS-induced sickness behavior, lethargy, and energy deficits within the first 24 h of treatment (Figure 3). This protective effect was, however, lost within the following 72 h of this low-dose LPS treatment regimen (Appendix A). Our short-term observations in LPA_5_^-/-^ mice might be due to lower peripheral TNFα or IL-1β (but unchanged IL-6) synthesis (Figure 1), both of these cytokines resembling classical inducers of sickness behavior in mice and humans [58]. Whether or not the partial recovery in response to repeated LPS injections is due to the development of endotoxin tolerance [59] is currently unclear. Of note, repeated injections of LPS trigger epigenetic modifications of microglia that result in activation of the mTOR/HIF-1α axis [60] and consequently lead to increased (aerobic) glycolysis [61], comparable to LPA-stimulated microglia observed in a previous [32] and in the present study (Figure 6).

In vitro, LPA-stimulated LPA_5_^-/-^ microglia showed reduced cyto-/chemokine synthesis, better mitochondrial fitness, and altered metabolic properties compared to wt cells, independent of whether microglia were exposed to 1 or 5 µM LPA (Figure 4, Figure 5 and Figure 6 and Appendix A, respectively). LPA_5_ was shown to play a pathogenic role during focal cerebral ischemia in mice with upregulated (mRNA and protein) expression of LPA_5_ in the ischemic core region, whereas LPA_5_ antagonism by TCLPA5 treatment significantly attenuated ischemic brain damage [22]. In addition, the authors demonstrated that siRNA-mediated LPA_5_ knockdown downregulated gene expression of pro-inflammatory cytokines in LPS-activated BV-2 microglia [22], comparable to our observations in primary LPA-treated LPA_5_^-/-^ microglia (Figure 4). In another model, ischemic brain also exhibited increased ATX activity, LPA concentrations, and LPA receptor expression, with LPA_5_ being the most pronounced [62]. These findings were accompanied by disrupted redox balance, BBB dysfunction, and reduced mitochondrial activity. All of these LPA-mediated pathological changes were reversed in response to an ATX and LPA pan-receptor inhibitor (BrP-LPA; [62]). These results are consistent with significantly higher maximal mitochondrial respiration and spare respiratory capacity in LPA_5_^-/-^ microglia compared to wt cells (Figure 5 and Appendix A). A detrimental role of LPA on mitochondrial function is further supported by the fact that heterozygous ATX knockout leads to improved mitochondrial energy homeostasis in brown adipose tissue of mice fed a high-fat, high-sucrose diet [63]. Similarly, a microarray-based approach in brown preadipocytes revealed that ATX-LPA signaling downregulates proteins involved in mitochondrial function and energy metabolism [64].

If LPA compromises mitochondrial function and oxidative phosphorylation, cells might be expected to increase glycolytic flux to meet their energy demand through aerobic glycolysis [65]. Indeed, treatment of wt microglia with LPA led to enhanced lactate secretion (Figure 6A), whereas this response was absent in LPA_5_^-/-^ cells, at least after 2 h, or even reversed after 24 h. The increase in the NADPH/NADP ratio (Figure 6B) is indicative for an increased metabolite flux through the oxidative branch of the pentose phosphate pathway (PPP). The antioxidant function of NADPH for thioredoxin activity and GSH recycling is well established. However, it becomes increasingly clear that this hexose monophosphate shunt-derived metabolite can also act as a pro-oxidant during O_2_^•−^ production by NADPH oxidases or NO generation by iNOS [66]. Our data suggest that LPA in wt microglia causes an increase in the NADPH/NADP ratio that is associated with elevated cellular GSH and NO production (Figure 6C,D). LPA signaling via LPA_5_ apparently plays a central role in these pathways, since these metabolic responses are either less pronounced or absent in LPA_5_^-/-^ microglia (Figure 6).

A recent study identified LPA as a modulator of the metabolic landscape in human pluripotent stem cells. The authors reported significantly increased relative amounts of several amino acids and glycolytic, TCA, and PPP intermediates [33], supporting our observations that LPA has the potential to induce metabolic rewiring in microglia ([32] and Figure 6). As for mechanistic pathway analysis, combined LPS/IFNγ treatment of microglia was shown to upregulate G6PDH expression, the first and rate-limiting enzyme of the PPP [67]. The same group showed that increased PPP activity feeds NADPH into NO and ROS synthesis, but also serves as a cofactor of glutathione reductase that converts GSSG back to GSH [67], which is consistent with our data in LPA-treated microglia from wt mice. In line, upregulated expression and activity of G6PDH in Parkinson’s disease (PD) models was accompanied by excessive NADPH and subsequent ROS production via NOX2 [68]. Knockdown or pharmacological inhibition of G6PDH ameliorated pro-inflammatory microglia polarization, ROS production, and NFκB activation [68]. NADPH-dependent NO synthesis by LPS/IFNγ-activated microglia relies exclusively on efficient glucose flux through the PPP [69]. Microglia use glucose as exclusive energy substrate to generate the superoxide anion radical via NOX2 that transfers electrons to molecular oxygen at the outer and oxidizing NADPH to NADP^+^ and H^+^ at the inner plasma membrane leaflet [70]. Thus, NADPH (generated via the PPP) can perform two seemingly opposing functions: (i) as an essential antioxidant cofactor of glutathione reductase and thioredoxin, and (ii) as a pro-oxidant trigger of “reductive stress” that leads to the formation of O_2_- and/or NO [66,71], highlighting the close link between redox regulation and immunometabolism [72]. Our observations that LPA_5_^-/-^ microglia have a lower NADPH/NADP ratio and produce less NO than wt microglia might suggest that this lipid-activated receptor is involved in the process termed “reductive stress”.

Although our study shows that LPA_5_ plays a critical role in neuroinflammation, there are also limitations: in our in vivo experiments, only male mice were used and it is unclear why LPA_5_^-/-^ mice were protected only during the first 24 h during the chronic LPS treatment regimen. For metabolic studies, it is noteworthy that the PPP is not the only source of NADPH, as substantial amounts are also generated via cytosolic and mitochondrial folate-dependent pathways [73] or by cytosolic isocitrate dehydrogenase and the malic enzyme [66]. Of note, NADPH generation during oxidation of methylene tetrahydrofolate to 10-formyl-tetrahydrofolate is also coupled to the cellular GSH/GSSG status [73]. Since we performed only enzymatic assays (and not stable isotope-labeled precursor flux analysis), we are unable to comment on the actual metabolic pathway(s) that generate NADPH. However, considering the strict glucose dependence of microglia for NO production [69], a substantial contribution of the PPP is likely.

Despite these potential shortcomings, our study clearly demonstrates that LPA_5_-mediated signaling cascades are centrally involved in the neuroinflammatory response. In this setting, LPA_5_^-/-^ animals and primary cells represent invaluable tools to verify and extend neurological in vivo and in vitro data obtained with pharmacological LPA_5_ antagonists.

## Figures and Tables

**Figure 1 cells-11-01071-f001:**
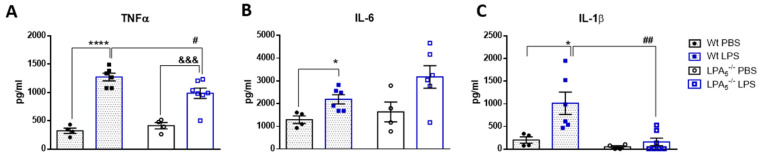
LPA_5_ deletion attenuates peripheral TNFα and IL-1β concentrations in LPS-injected mice. Wt and LPA_5_^-/-^ mice were injected i.p. with PBS (*n* = 4) or LPS (5 mg/kg; *n* = 6). After 24 h, the animals were sacrificed, blood was collected and serum was isolated. The concentrations of (**A**) TNFα, (**B**) IL-6, and (**C**) IL-1β were quantified using ELISAs. Values are expressed as mean ± SEM, * *p* < 0.05, **** *p* < 0.0001 compared to wt PBS control; &&& *p* < 0.001 compared to LPA_5_^-/-^ PBS control; # *p* < 0.05, ## *p* < 0.01 compared to LPS-treated wt mice; two-way ANOVA with Bonferroni correction).

**Figure 2 cells-11-01071-f002:**
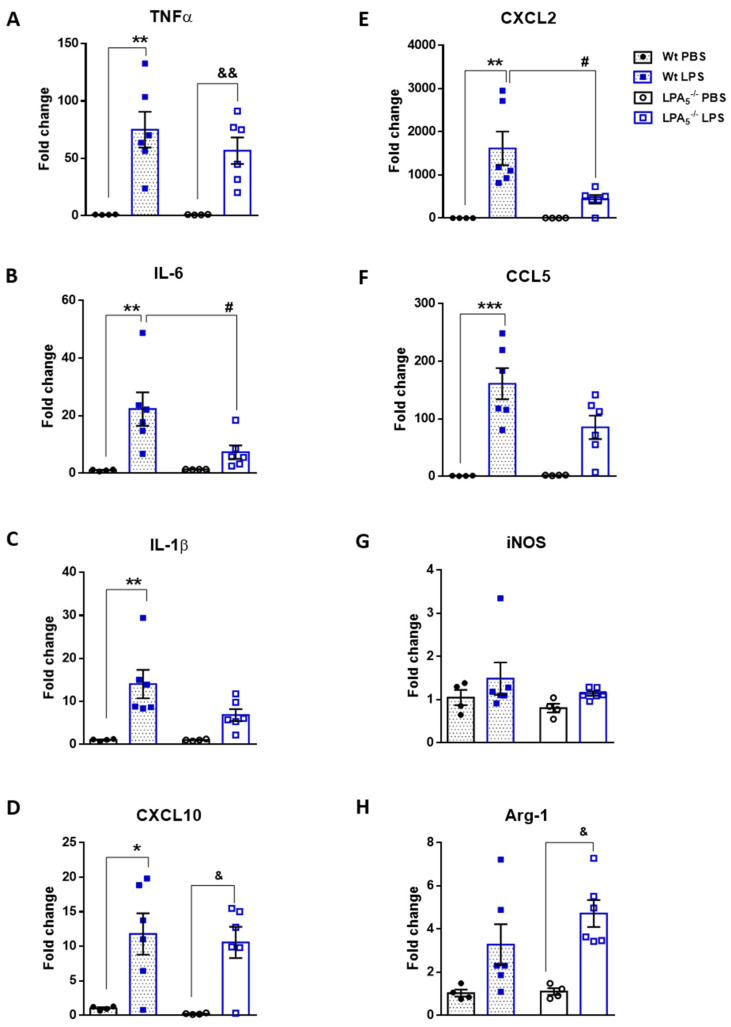
LPA_5_ deletion attenuates expression of pro-inflammatory genes in mouse brain. Wt and LPA_5_^-/-^ mice were injected i.p. with PBS (*n* = 4) or LPS (5 mg/kg; *n* = 6). After 24 h, the animals were sacrificed and perfused. Brains were processed for RNA isolation and gene expression of (**A**) TNFα, (**B**) IL-6, (**C**) IL-1β, (**D**) CXCL10, (**E**) CXCL2, (**F**) CCL5, (**G**) iNOS, and (**H**) Arg-1 was evaluated by qPCR. Hypoxanthine-guanine phosphoribosyltransferase (HPRT) was used as housekeeping gene. Expression was calculated using the 2^−ΔΔCt^ method. Results are presented as mean values ± SEM, * *p* < 0.05, ** *p* < 0.01, *** *p* < 0.001 compared to wt PBS control; & *p* < 0.05, && *p* < 0.01 compared to LPA_5_^-/-^ PBS control; # *p* < 0.05 compared to LPS-treated wt mice; two-way ANOVA with Bonferroni correction).

**Figure 3 cells-11-01071-f003:**
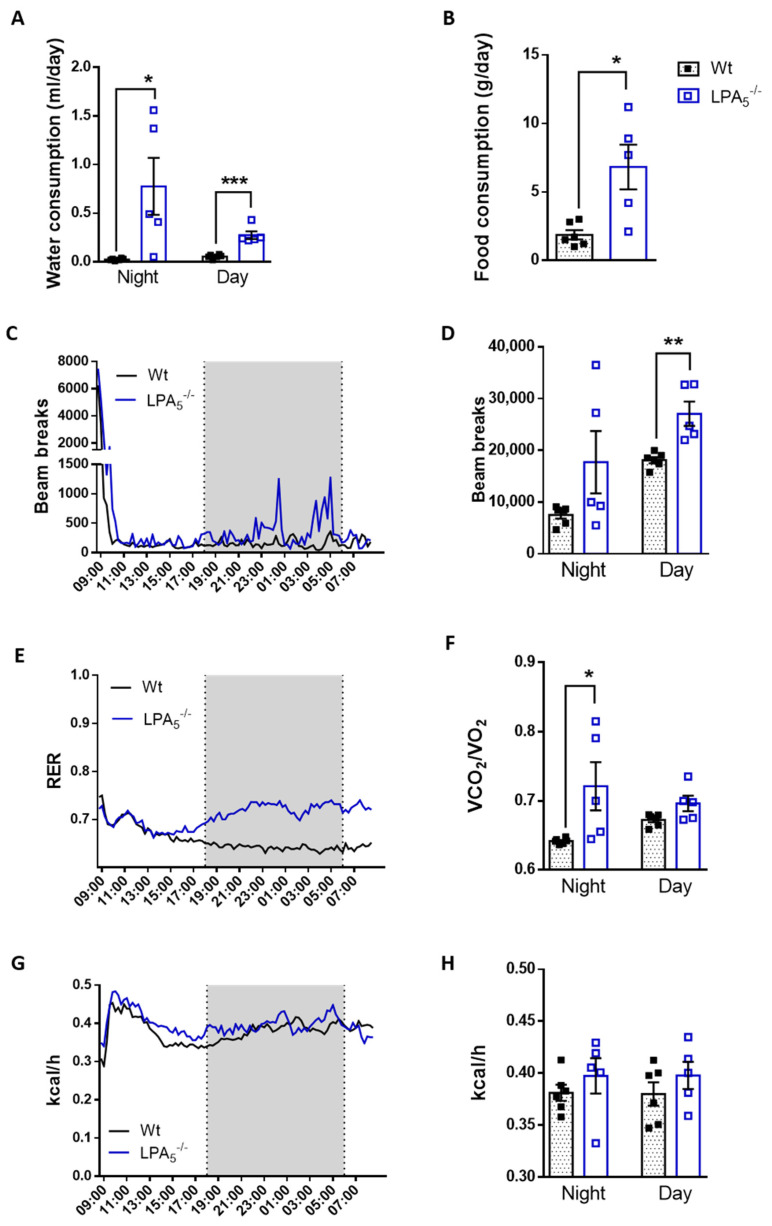
Improved metabolic performance of LPA_5_^-/-^ mice after initial LPS treatment. Short-term metabolic cage readouts 24 h after the first LPS injection are shown. Wt and LPA_5_^-/-^ mice were housed at room temperature in metabolic cages with free access to chow diet and water. Mice were injected daily with LPS (i.p., 1.4 mg/kg body weight) for 4 d. (**A**) water consumption, (**B**) food intake, (**C**) real-time locomotor activity, (**D**) mean locomotor activity, (**E**) real-time measurement of respiratory exchange ratio (RER), (**F**) mean RER, (**G**) real-time energy expenditure (EE) measurement, and (**H**) mean EE. (Data represent as mean values ± SEM for wt (*n* = 6) and LPA_5_^-/-^ (*n* = 5) mice. Significance was calculated by Student’s *t*-test. * *p* < 0.05, **, *p* < 0.01, *** *p* < 0.001 compared to LPS-treated wt mice.

**Figure 4 cells-11-01071-f004:**
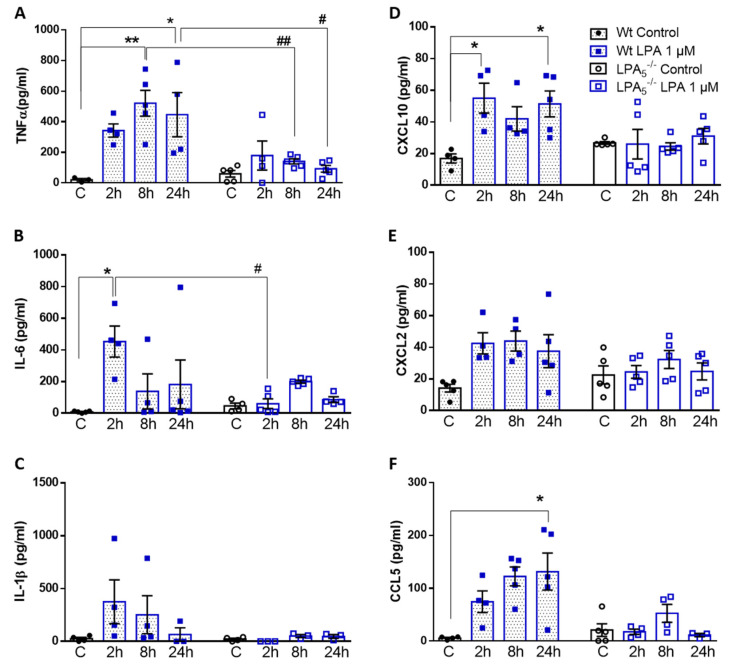
LPA_5_ regulates LPA-induced secretion of cyto-/chemokines in primary microglia. Wt and LPA5^-/-^ cells were treated in the absence (**C**) or presence of LPA (1 µM) for the indicated times. Supernatants were collected and (**A**–**C**) cytokine (TNFα, IL-6, and IL-1β) and (**D**–**F**) chemokine (CXCL10, CXCL2, and CCL5) concentrations were quantified by ELISA. Values are expressed as mean ± SEM of five independent experiments. * *p* < 0.05, ** *p* < 0.01 compared to wt control; # *p* < 0.05, ## *p* < 0.01 for LPA_5_^-/-^ compared to wt cells (two-way ANOVA with Bonferroni correction).

**Figure 5 cells-11-01071-f005:**
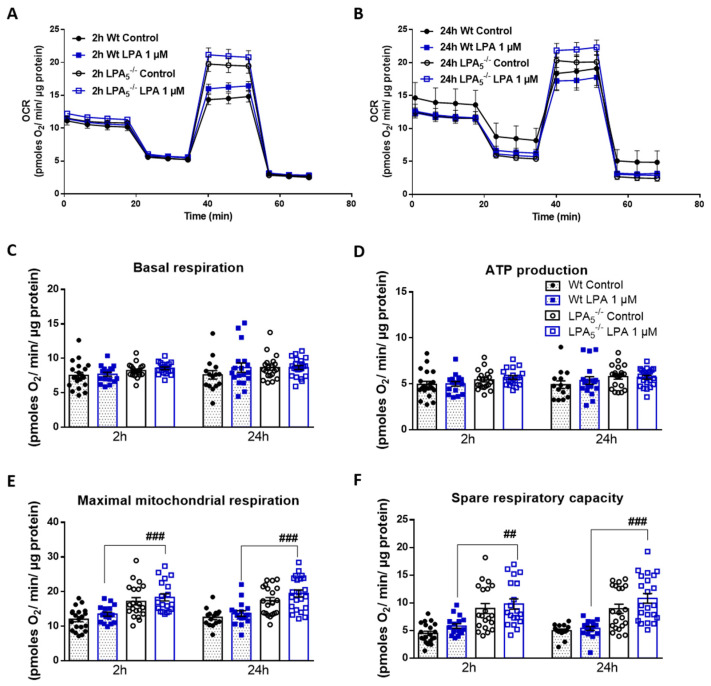
LPA_5_ regulates mitochondrial respiration in primary microglia. (**A**,**B**) Oxygen consumption rate (OCR) in the absence and presence of LPA (1 µM) for 2 or 24 h was measured using the XF Cell Mito Stress Test. Primary microglia isolated from wt and LPA5^-/-^ mice were treated with 2 μM oligomycin, 1.75 μM FCCP, and 2.5 μM antimycin A in XF assay medium to assess mitochondrial function parameters. Bar graphs show (**C**) basal mitochondrial respiration, (**D**) maximal mitochondrial respiration, (**E**) ATP linked respiration, and (**F**) spare respiratory capacity. Values are expressed as mean ± SEM of three independent experiments. ## *p* < 0.01, ### *p* < 0.001 LPA_5_^-/-^ compared to wt cells; two-way ANOVA with Bonferroni correction.

**Figure 6 cells-11-01071-f006:**
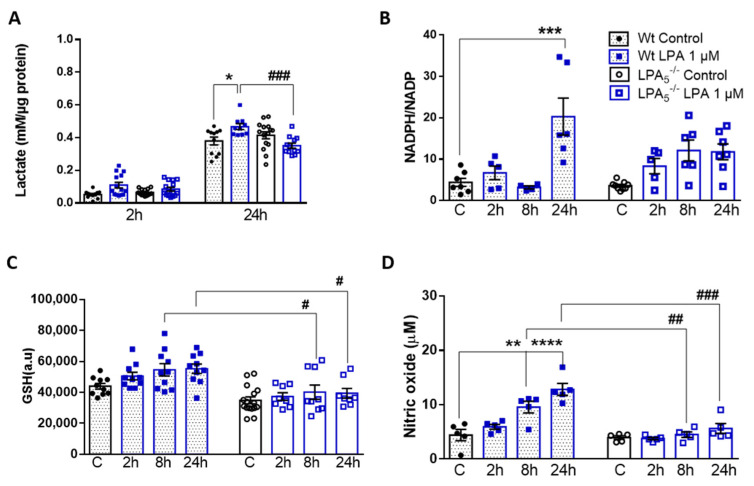
LPA_5_ regulates extracellular lactate content, NADPH/NADP ratio, glutathione concentration, and NO production in primary microglia. Wt and LPA_5_^-/-^ cells were cultivated in serum-free medium in the absence or presence of LPA (1 µM). (**A**) Lactate content was measured by EnzyChrom™ Glycolysis Assay Kit. (**B**) NADPH/NADP ratio was measured using the NADP/NADPH assay kit (Abcam). (**C**) Glutathione (GSH) concentration was quantified with the GSH-Glo Assay kit. (**D**) The production of NO was determined by measuring the total nitrate concentration in the supernatants. Results are presented as mean values ± SEM of three independent experiments. * *p* < 0.05, ** *p* < 0.01, *** *p* < 0.001, **** *p* < 0.0001 compared to wt controls, # *p* < 0.05, ## *p* < 0.01, ### *p* < 0.001 for LPA_5_^-/-^ compared to wt cells, two-way ANOVA with Bonferroni correction.

**Table 1 cells-11-01071-t001:** Primers used for qPCR experiments during the present study.

**Gene**	**Company**	**Catalogue Number**
iNOS	Qiagen	QT00100275
HPRT	Qiagen	QT00166768
**Gene**	**Company**	**Forward/Reverse Primers**
Arg-1	Invitrogen	F: TGGCTTGCGAGACGTAGAC R: GCTCAGGTGAATCGGCCTTTT
TNFα	Invitrogen	F: ACTTCGGGGTGATCGGTCC R: GGCTACAGGCTTGTCACTCG
IL6	Invitrogen	F: TGTTCTCTGGGAAATCGTGGA R: CAAGTGCATCATCGTTGTTCAT
IL1β	Invitrogen	F: CTCTCCACCTCAATGGACAGA R: CGTTGCTTGGTTCTCCTTGT
CXCL10	Invitrogen	F: TTCTGCCTCATCCTGCTG R: AGACATCTCTGCTCATCATTC
CXCL2	Invitrogen	F: AGTGAACTGCGCTGTCAATG R: GCCCTTGAGAGTGGCTATGA
CCL5	Invitrogen	F: GCTGCTTTGCCTACCTCTCC R: TCGAGTGACAAACACGACTGC

## Data Availability

The data presented in this study are available on reasonable request from the corresponding author. Reagents and detailed methods of all procedures are provided in the “Materials and Methods” of this manuscript or cited accordingly.

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
