# Peer review of "Lysophosphatidic Acid Receptor 5 (LPA_5_) Knockout Ameliorates the Neuroinflammatory Response In Vivo and Modifies the Inflammatory and Metabolic Landscape of Primary Microglia In Vitro"

_cells, 2022, doi:10.3390/cells11071071_

Round 1

Reviewer 1 Report

The manuscript by Joshi et al investigates the role of LPA5 in the neuroinflammation in a mouse endotoxemia model using ELISA, qRT-PCR, along with metabolic parameters analysis. Analysis is followed up using in vitro LPA-activated primary microglial cells. Overall, the manuscript looks very interesting and the whole study seems to be acceptable. However, I have some concerns which need to be addressed.

  1. In previous study, the role of LPA5 in neuroinflammation is already demonstrated in endotoxemia mouse model (Joshi et al., 2021, IJMS) and LPA-induced activated primary microglia (Plastira et al., 2016, J of Neuroinflammation) using LPA5 Please discuss the rationale and novelty of this study.
  2. The LPS-induced endotoxemia mouse model is quite severe (1.4 mg/kg i.p. daily for 4 days). Mice are probably close to being lethargic. What is the rationale to use such a high dose and so many days? Systemic inflammation is sufficient with a single dose as it sustains over several days.
  3. Please discuss the clinical importance of this study.
  4. Make a consistent writing pattern for dose of LPS (Lines 105 and 115). Please mention the route of administration (line 115).
  5. In figure 2 legend, please remove the repetition information (number of mice).
  6. Please mention the number of samples using in in vitro experiment i.e. Figure 4, 5, and 6.
  7. In figure 3, most of the metabolic performance parameters were measured at day and night. However, no specification of time for food consumption measurement. Please make a consistent data presentation.
  8. Correct typo LPA5 (line 318)

Author Response

Response to Reviewer #1

Comments and Suggestions for Authors

The manuscript by Joshi et al investigates the role of LPA5 in the neuroinflammation in a mouse endotoxemia model using ELISA, qRT-PCR, along with metabolic parameters analysis. Analysis is followed up using in vitro LPA-activated primary microglial cells. Overall, the manuscript looks very interesting and the whole study seems to be acceptable. However, I have some concerns which need to be addressed.

RESPONSE

We thank the reviewer for the time and the constructive and valuable suggestions for revision. All of the recommendations were followed.

  1. In previous study, the role of LPA5 in neuroinflammation is already demonstrated in endotoxemia mouse model (Joshi et al., 2021, IJMS) and LPA-induced activated primary microglia (Plastira et al., 2016, J of Neuroinflammation) using LPA5 Please discuss the rationale and novelty of this study.

RESPONSE

Thanks for the suggestion, the rationale was added in the last paragraph of the Introduction; lines 84-86 track version) and the novelty of our findings (including potential for clinical application) is briefly summarized in the first paragraph of the Discussion (lines 609-625).

  1. The LPS-induced endotoxemia mouse model is quite severe (1.4 mg/kg i.p. daily for 4 days). Mice are probably close to being lethargic. What is the rationale to use such a high dose and so many days? Systemic inflammation is sufficient with a single dose as it sustains over several days.

RESPONSE

This chronic lower-dose regimen was chosen to monitor animal behavior over an extended period of time. According to our experience LPS at 5 mg/kg body weight induces severe animal suffering between 18-24 h (approx. 1d after injection), therefore we have chosen the lower dose chronic treatment from which mice recover. This is now clearly mentioned in the M&M section (subheading Indirect Calorimetry). In addition, we have added a study be Chen and colleagues (new Reference 35) that demonstrated global microglia activation in C57BL/6 mice using a comparable dosing regimen (1 mg LPS/kg daily for 4 days).

  1. Please discuss the clinical importance of this study.

RESPONSE

The potential clinical importance of LPA5 antagonism is briefly covered in the first paragraph of the discussion. Despite promising results of LPA5 inhibition in rodent models we also point out that findings in animal models do not necessarily reproduce in the human system (lines 615-625).

  1. Make a consistent writing pattern for dose of LPS (Lines 105 and 115). Please mention the route of administration (line 115).

RESPONSE

This was amended (i.p.).

  1. In figure 2 legend, please remove the repetition information (number of mice).

RESPONSE

This was corrected.

  1. Please mention the number of samples using in in vitro experiment i.e. Figure 4, 5, and 6.

RESPONSE

This information is now included in the Figure legends of the revised version (and also in the Supplement).

  1. In figure 3, most of the metabolic performance parameters were measured at day and night. However, no specification of time for food consumption measurement. Please make a consistent data presentation.

RESPONSE

This is an important point. The physical construction of the metabolic cages with the integrated infrared light-beam activity monitor requires that the lowest part of the food container is above the beam grid to prevent interference with the light beams and enable reliable measurement of locomotor activity. In this case, the construction of the food container itself requires the animals to stand on their hind legs and reach for the food. Since this may be too challenging for some animals after the LPS injections, we decided to place food pellets directly in the cage and measure food consumption manually. During the adaption phase, cumulative food consumption was divided by the days of the adaption phase to measure average food consumption. During the LPS treatment, food consumption was measured manually every day just before the LPS injection. This is clearly described in the M&M section of the revised manuscript (subheading Indirect Calorimetry, lines 128-134).

  1. Correct typo LPA5 (line 318)

RESPONSE

This was corrected.

Reviewer 2 Report

In this manuscript, Joshi et al. examined the role of the LPA5 in the neuroinflammatory response using LPA5 KO mice. They found that LPS-induced increases in TNFα and IL-1β levels in the serum and gene expression of IL-6, CXCL2, and CCL5 in the brain are decreased in LPA5 KO mice. They also found that the LPA responses (cyto-/chemokines secretion, lactate, NADPH, and NO synthesis, and mitochondrial respiration) are changed in LPA5-deficient microglia. Additionally, they show that PLA5 mice show improved metabolic performance after short-term low-dose LPS treatment. Their findings are important for understanding the mechanism of inflammation response. Overall the data are convincing. I therefore recommend publication of this paper in Cells, after the authors address the following points

Some statistical analyses are inappropriate (Figures 1, 2, 6). For multiple comparisons among means, ANOVA with appropriate post hoc analysis should be used rather than student t-test.

Minor comments

1) There are several different font characters in the manuscript.

2) ”n=6” should be “n = 6” (page 6, line 247). “37 ℃” should be “37℃” (page 4, line 152, 157). “p<0.01” should be ““p < 0.01” (page 7, line 253; page 8, line 286). * p < 0.05**p < 0.01” should be “* p < 0.05, **p < 0.01” ((page9, lone 305), etc.

The authors should carefully check and correct these throughout the manuscript.

Author Response

Response to Reviewer #2

Comments and Suggestions for Authors

In this manuscript, Joshi et al. examined the role of the LPA5 in the neuroinflammatory response using LPA5 KO mice. They found that LPS-induced increases in TNFα and IL-1β levels in the serum and gene expression of IL-6, CXCL2, and CCL5 in the brain are decreased in LPA5 KO mice. They also found that the LPA responses (cyto-/chemokines secretion, lactate, NADPH, and NO synthesis, and mitochondrial respiration) are changed in LPA5-deficient microglia. Additionally, they show that PLA5 mice show improved metabolic performance after short-term low-dose LPS treatment. Their findings are important for understanding the mechanism of inflammation response. Overall the data are convincing. I therefore recommend publication of this paper in Cells, after the authors address the following points

RESPONSE

We thank the reviewer for the time and the constructive and valuable suggestions for revision. All of the recommendations were followed.

Some statistical analyses are inappropriate (Figures 1, 2, 6). For multiple comparisons among means, ANOVA with appropriate post hoc analysis should be used rather than student t-test.

RESPONSE

This is a valid point and two-way ANOVA was used as suggested. This led to the loss of statistical significance in some instances and this was amended in the test (e.g. in Fig. 2 only IL-6 and CXCL2 are significantly lower in LPA5-/- animals; line 420 track version).

Minor comments

1) There are several different font characters in the manuscript.

2) ”n=6” should be “n = 6” (page 6, line 247). “37 ℃” should be “37℃” (page 4, line 152, 157). “p<0.01” should be ““p < 0.01” (page 7, line 253; page 8, line 286). * p < 0.05**p < 0.01” should be “* p < 0.05, **p < 0.01” ((page9, lone 305), etc.

The authors should carefully check and correct these throughout the manuscript.

RESPONSE

Thank you for drawing our attention to these typos, all of them were corrected.

Round 2

Reviewer 1 Report

Everything looks good.